# Detection of *Babesia odocoilei* in *Ixodes scapularis* Ticks Collected from Songbirds in Ontario and Quebec, Canada

**DOI:** 10.3390/pathogens9100781

**Published:** 2020-09-24

**Authors:** John D. Scott, Emily L. Pascoe, Muhammad S. Sajid, Janet E. Foley

**Affiliations:** Department of Medicine and Epidemiology, School of Veterinary Medicine, University of California Davis, Davis, CA 95616, USA; elpascoe@ucdavis.edu (E.L.P.); mssajid@ucdavis.edu (M.S.S.); jefoley@ucdavis.edu (J.E.F.)

**Keywords:** *Babesia odocoilei*, piroplasm, human babesiosis, songbirds, ticks, *Ixodes scapularis*, parasitism, Canada

## Abstract

Songbirds widely disperse ticks that carry a diversity of pathogens, some of which are pathogenic to humans. Among ticks commonly removed from songbirds, the blacklegged tick, *Ixodes scapularis*, can harbor any combination of nine zoonotic pathogens, including *Babesia* species. From May through September 2019, a total 157 ticks were collected from 93 songbirds of 29 species in the Canadian provinces of Ontario and Québec. PCR testing for the 18S gene of *Babesia* species detected *Babesia odocoilei* in 12.63% of *I. scapularis* nymphs parasitizing songbirds in Ontario and Québec; none of the relatively small numbers of *Ixodes muris*, *Ixodes brunneus*, or *Haemaphysalis leporispalustris* were PCR-positive. For ticks at each site, the prevalence of *B. odocoilei* was 16.67% in Ontario and 8.89% and 5.26% in Québec. Of 31 live, engorged *I. scapularis* larvae and nymphs held to molt, 25 ticks completed the molt; five of these molted ticks were positive for *B. odocoilei*. PCR-positive ticks were collected from six bird species—namely, Common Yellowthroat, Swainson’s Thrush, Veery, House Wren, Baltimore Oriole, and American Robin. Phylogenetic analysis documented the close relationship of *B. odocoilei* to *Babesia canis canis* and *Babesia divergens*, the latter a known pathogen to humans. For the first time in Canada, we confirm the transstadial passage of *B. odocoilei* in *I. scapularis* molting from larvae to nymphs. A novel host record reveals *I. scapularis* on a Palm Warbler. Our findings show that *B. odocoilei* is present in all mobile life stages of *I. scapularis*, and it is widely dispersed by songbirds in Ontario and Québec.

## 1. Introduction

Ticks carry a wide spectrum of disease-causing microorganisms that cycle between vectors and hosts, including humans. Some of these ectoparasites are laden with tick-borne zoonotic pathogens, including *Babesia* spp. (Apicomplexa: Piroplasmida: Babesiidae). These malaria-like piroplasms, which invade red blood cells, were first discovered by the Romanian researcher Victor Babes in 1888 [1,2,3]. The first description of human babesiosis was a fatal case in a splenectomized farmer in Croatia [4]. Although ticks were identified as vectors shortly after the discovery of the pathogen [5], more recently, this apicomplexan microorganism has also been found to be transmitted to humans by blood transfusion [6,7], by organ transplantation [8], and by maternal–fetal transmission [9,10]. *Babesia* species are typically host-specific and, globally, they infect a wide range of vertebrate hosts. These *Babesia* spp. often result in reduced fitness and mortality in wildlife and domestic animals, producing economic losses with low productivity and added mortality. Epidemiologically, they also cause morbidity and mortality in humans worldwide [2,3,4,11,12,13,14,15,16]. Human babesiosis is reported in Canada and the United States, sometimes with fatal outcomes [11,17]. *Babesia* species that bring about pathogenicity in humans include *B. microti, B. crassa, B. duncani, B. venatorum, B. divergens*, *Babesia divergens*-like MO-1, *Babesia* sp. KO-1, *Babesia* sp. XXB/HangZhou, and *Babesia* spp. CA1, CA3, and CA4 [11,17,18,19].

Migratory songbirds (Passeriformes) disperse ticks and tick-borne pathogens across regional and transcontinental areas. These migrants occasionally support the establishment of ticks and pathogens in new foci [20,21]. Some songbirds, such as the American Robin, are reservoirs of the Lyme disease bacterium, *Borrelia burgdorferi* sensu lato [22]; however, passerine species are not known to be reservoir-competent hosts of *Babesia* spp. Even if birds are refractory to infection with a tick-borne pathogen, they can host ticks that are already infected with disease-producing microbes. For example, the blacklegged tick, *Ixodes scapularis* (Acari: Ixodidae), an important human-biting tick in eastern and central North America, can harbor any combination of nine different zoonotic pathogens, including *Babesia* species [23]. A nymphal *I. scapularis*, which parasitized a spring-migrating Veery in southeastern New York state, was triple co-infected with *B. burgdorferi* sensu lato, *Babesia microti*, and *Anaplasma phagocytophilum* [24].

Certain cervids (i.e., caribou, deer, elk) are reservoirs of *Babesia odocoilei*, while *I. scapularis* ticks are vectors of this babesial infection. In many parts of eastern and central North America, *I. scapularis* ticks and cervine hosts co-habitat in nature to maintain *B. odocoilei*. Passerine birds are common incidental hosts of *I. scapularis* larvae and nymphs, and occasionally, these ticks are infected with *B. odocoilei*. Based on current knowledge, cervids and *I. scapularis* form a collective hub for the enzootic transmission cycle of *B. odocoilei*. Although *B. odocoilei* has been investigated as a pathogen of cervids, it has not been examined closely as a potential source of human babesiosis. 

The aim of this study was to detect and identify, to species, any *Babesia* DNA in ticks collected from migratory songbirds in two provinces in Canada. 

## 2. Materials and Methods

### 2.1. Collection of Ticks from Songbirds

Bird banders and wildlife rehabilitators collected ixodid ticks from songbirds captured at five locations (Figure 1) between 23 April and 30 September 2019. They removed ticks using fine-pointed hardened stainless steel, #5, superfine-tipped forceps (BioQuip Products, Rancho Dominguez, CA, USA). There were three collection sites in Ontario (Ruthven Park, Long Point, and Toronto) and two in Québec (Ste-Anne-de-Bellevue and Montée Biggar). These technicians and collectors placed live ticks into transparent 8.5 mL, 15.7 mm × 75 mm, round-bottomed polypropylene tubes (Sarstedt, Montréal, Québec, Canada); each tube contained ticks from a single host. In order to prevent ticks from escaping and provide ventilation, a 3 cm piece of tulle netting was placed over the mouth of the tube, and a push cap, which had a 7 mm hole, was inserted into the tube opening. All ticks were promptly mailed in a double-zipper plastic bag with a moistened paper towel to the laboratory (JDS) for identification. Partially and fully engorged larval and nymphal ticks were held to molt to the next developmental life stage at a long-day photoperiod of 16:8 (L:D) h at room temperature in humidity from 90% to 95%. For the statistical analyses, molted ticks were designated the life stage at the time of collection. Taxonomic keys and descriptions were utilized for identification [25,26,27,28,29]. 

### 2.2. DNA Extraction and PCR

Unfed ticks were subjected to an ammonium hydroxide DNA extraction protocol as described previously [30], whereas DNA from fed ticks was extracted using the Qiagen DNeasy Blood and Tissue Kit (Qiagen, Valencia, CA, USA) following the manufacturer’s protocol for animal tissue. Ticks were initially pooled according to bird host; however, upon discovering the relatively high prevalence, we decided to process ticks individually. DNA was stored at −20 °C until PCR was performed. Conventional PCR was used to detect the 18S ribosomal RNA (rRNA) gene of the genus *Babesia*, with 25 µL reaction mixes containing 2.5 µL each of 5 µM primers BJ1 (5′-GTC-TTG-TAA-TTG-GAA-TGA-TGG-3′) and BN2 (5′-TAG-TTT-ATG-GTT-AGG-ACT-ACG-3′) [31], 5 µL nuclease-free water, 12.5 µL Green Go Taq (Promega, Madison, WI, USA), and 2.5 µL of DNA at 8–10 ng/µL. PCR was performed under thermal cycling conditions of initial denaturation at 94 °C for 10 min, 35 cycles of 94 °C for 1 min, 55 °C for 1 min, and 72 °C for 2 min with a final extension at 72 °C for 5 min, and then held at 4 °C. Amplified DNA was visualized with UV transillumination of a 1% agarose gel containing GelStar nucleic acid stain (Lonza, Rockland, ME, USA). Amplicons of 400–500 base pairs (bp) were excised from the gel and prepared for DNA sequencing using a gel extraction kit (QIAamp DNA Kit, Qiagen, Valencia, CA, USA). 

### 2.3. DNA Sequencing and Phylogenetic Analysis

DNA sequencing was performed at UC Davis DNA Sequencing (College of Biological Sciences at University of California Davis, CA, USA) using the Big Dye Terminator cycle sequencing kit (Applied Biosystems, Foster City, CA, USA) and PCR primers. Sequences were compared to those published in GenBank using the BLAST database search program (https://blast.ncbi.nlm.nih.gov/Blast.cgi#_blank). In addition, sequences were manually corrected for ambiguous base calls and to remove end-reading errors. All sequences were trimmed to the same length, and the MUSCLE algorithm performed the sequence alignments [32]. DNA sequences from many *Babesia* species comprised *B. odocoilei*, *B. bovis*, *B. conradae*, *B. divergens*, *B. duncani*, *B. canis canis*, *B. gibsoni*, *B. microti*, and *B. vulpes* from GenBank for inclusion in the phylogenetic tree. Phylogeny was resolved using the maximum likelihood method in MEGA 10.0.5 [33] using the Hasegawa-Kishino-Yano with gamma distribution as determined by jModeltest 2.1.10 [34]. Bootstrapping was performed based on 1000 pseudoreplicate datasets generated from the original sequence alignments.

## 3. Results

### 3.1. Tick Collection

Overall, 157 ixodid ticks were collected from 93 songbirds of 29 bird species in 2019 at five sites in Ontario (ON) and Québec (QC) (Table 1). Species of ticks included *Haemaphysalis leporispalustris* (rabbit tick) (n = 6), *Ixodes brunneus* (bird tick) (n = 4), *Ixodes muris* (mouse tick) (n = 21), and *I. scapularis* (n = 126). The majority of ticks were nymphs (n = 134), followed by larvae (n = 19), and only four adults. Of 31 engorged ticks that were held to molt, nine successfully molted from larvae to nymphs (six *I. muris* and three *I. scapularis*), eight *I. scapularis* molted from nymph to adult female, seven *I. scapularis* molted from nymph to adult male, and one female *I. muris* laid a clutch of eggs, which hatched to larvae. Two *I. scapularis* larvae were collected from a Swainson’s Thrush on 31 May 2019 at Long Point, ON. These larvae molted to nymphs and tested positive for *B. odocoilei*; they constitute the first transstadial passage of *B. odocoilei*-positive larvae molting to nymphs in Canada.

The highest tick burden occurred at Montée Biggar, QC, with an average of 2.44 ticks per individual bird, followed by Ruthven Park, ON (1.84), Sainte-Anne-de-Bellevue, QC (1.36), Long Point, ON (1.33), and Toronto, ON (1.0). *Ixodes scapularis* was the only tick species found at all sampled locations, and all four tick species were present at Ruthven Park and Sainte-Anne-de-Bellevue.

The mean tick burden across all bird species was 1.69 ticks per individual with American Robins having the greatest tick burden with an average of 4.0 ticks (Table 1). Veeries, House Wrens, Winter Wrens, Hermit Thrushes, and Swainson’s Thrushes each had an average burden of 2.0–2.7 ticks per individual. Tick burdens in all other tick species were lower. Of note, *I. scapularis* was collected from a Palm Warbler at Long Point, Ontario on 14 May 2019, and it denotes the first report of this host record. *Ixodes muris* were recorded on 12 bird species, and *I. brunneus* and *H. leporispalustris* were found on three bird species each.

### 3.2. Babesia Detection

At least 12 ticks, all of which were *I. scapularis*, tested positive for *Babesia*, including a pool of four ticks collected from a Common Yellowthroat, a pool of five from a Common Yellowthroat, and a pool of seven from an American Robin. In general, the prevalence of *Babesia* in ticks was at least 8.89% (12/135), and it was highest at Ruthven Park (16.67%), followed by 8.89% at Sainte-Anne-de-Bellevue, and 5.26% at Montée Biggar (Table 2). There were no positive samples from the other two locations. The highest prevalence of *Babesia* in bird-feeding ticks was Swainson’s Thrush (30.00%, 3/10 tested pools), Baltimore Oriole (25.0%), American Robin (16.67%, 1/6 tested pools), Common Yellowthroat (15.63%, 5/32 tested pools), and House Wren (14.29%). DNA sequencing confirmed all positive samples as *B. odocoilei*; sequences were 99.18–100% similar, and all samples were confirmed as *B. odocoilei*. With the exception of four nymphs pooled from a Common Yellowthroat (99.59%) and a single nymph from a Common Yellowthroat both from Ruthven Park, amplicons were 100% similar to three published *B. odocoilei*. The closest related species was *B. canis canis* (93.33–93.82% similarity to reference samples), which differed from *B. odocoilei* by 18 base pairs. Upon further analysis, we noticed a 92.88% similarity between sequences of *B. divergens* and *B. odocoilei* in the phylogenic tree (Figure 2).

The prevalence for both larvae and nymphs combined revealed that 12 (11.54%) of 104 *I. scapularis* ticks were positive for *B. odocoilei*. PCR-positive nymphs were collected between 5 May and 20 August, and the larvae were collected on 31 May. These larval and nymphal ticks included two larvae that were held to molt, and were tested as nymphs. The GenBank accession numbers for *B. odocoilei*, and their bird-tick-pathogen associations are listed in Table 3.

The sequences from tick DNA samples were all identical, except for CN19-33B and pool CN19-47B, which differed each by a single nucleotide from the other sequences. Therefore, these two sequences were on separate branches within the same clade of *B. odocoilei* sequences.

## 4. Discussion

As vectors, *I. scapularis* ticks play an integral role in the enzootic transmission cycle of *B. odocoilei*. Of medical significance, these arthropod vectors, which are commonly infected with pathogenic microbes, bite humans. Since *I. scapularis* larvae and nymphs parasitize birds, they are transported continental distances during spring and fall migration. Geographically, avian hosts can transport *B. odocoilei*-infected *I. scapularis* larvae and nymphs hundreds of kilometers during the migratory flight. This study highlights the major role that songbirds play in the dispersal of bird-feeding ticks and their associated pathogens, especially *B. odocoilei*. We show that the prevalence of *B. odocoilei* transported by wild birds is significant.

At the northern extent of migratory avian flyways that cover most of the western hemisphere [35], Canada experiences ticks and tick-transmitted pathogens from birds each spring from as far south as equatorial South America [36,37,38,39,40]. While Canadians contend with numerous tick-borne diseases, recent bird–tick–pathogen evidence reveals *I. scapularis* nymphs as far north and as far west as northern Alberta [41], and some of these ticks were infected with *B. burgdorferi* sensu lato [41]. As well, in Atlantic Canada, researchers have reported songbird-transported *I. scapularis* nymphs infected with *B. burgdorferi* sensu lato as far north as the province of Newfoundland and Labrador [41]. Here, we show that at least 29 migrating and locally active bird species carry multiple tick species across a wide area of southern Canada [42], and at least 12.63% of the nymphal *I. scapularis* infesting these avian hosts bring with them the apicomplexan pathogen *B. odocoilei*. 

Four species of ticks were removed from songbirds, including *I. muris* and *H. leporispalustris*, which commonly infest small mammals and birds. The bird tick, *I. brunneus*, is exclusively on birds [28], whereas *H. leporispalustris*, *I. muris*, and *I. scapularis* parasitize both avian and mammalian hosts. The latter two tick species bite humans, and *I. scapularis* is the predominant vector of pathogens associated with borreliosis, anaplasmosis, and babesiosis [23]. Biologically, *I. scapularis* has at least 150 vertebrate hosts (i.e., avian, mammalian, reptilian), and of these hosts, *I. scapularis* parasitize at least 82 different bird species [26,42,43]. The most commonly infested birds in our study were American Robins, Veeries, House Wrens, Winter Wrens, Hermit Thrushes, and Swainson’s Thrushes, all with an average of 2–4 ticks per individual bird. These ground-foraging species scratch and pick through the leaf litter, which facilitates bird parasitism by ectoparasites, especially host-seeking ticks. During fall migration, all ticks from short- and long-distance songbirds originated in Canada. Notably, we report a new host record of *I. scapularis* parasitizing a Palm Warbler. 

With the exception of Blue Jays, all of the passerines captured are migratory and support the transportation of *I. scapularis* northward during spring migration. The bird species that carried ticks with the highest prevalence of *B. odocoilei* (Figure 3) were Common Yellowthroat, Swainson’s Thrush, Veery, House Wren, Baltimore Oriole, and American Robin. Our findings revealed that up to 25% of orioles were parasitized. 

*Babesia odocoilei* is a recently described malaria-like microorganism that was originally reported as apathogenic in cervine hosts [44]. Now, it is known to cause death associated with high levels of parasitemia in cervids (Artiodactylia: Cervidae). This cervid family includes white-tailed deer (*Odocoileus virginianus*), American elk (*Cervus elaphus canadensis*), and caribou (*Rangifer terandus caribou*) [45,46]. Mortality has been noted in captive and wild cervids in various parts of North America.

Based on phylogenetic analysis of the 18S rRNA gene, *B. divergens* is the closest known genetic relative to *B. odocoilei*; the former is a human pathogen [11,19]. Even though the *B. odocoilei* amplicons in this study formed a close-knitted clade, they were widely distributed geographically. For instance, the direct flight path between Site 1 and Site 5 is 607 km. Since serological assays are not able to discriminate between certain babesial strains and species [31,47], DNA sequence analysis of PCR amplicons is needed to confirm the identity of any given *Babesia* species. 

To date, *I. scapularis* is the only vector known to harbor and transmit *B. odocoilei*. This particular piroplasm has been detected in *I. scapularis* ticks in multiple states of the U.S.A., including Indiana [48,49], Maine [49], Wisconsin [49], Michigan [50], New York [51], and Pennsylvania [49,52]. Prior to the present study, *B. odocoilei*-positive *I. scapularis* have been reported parasitizing songbirds in southwestern Ontario [53,54]. In northeastern North America, a *B. odocoilei*-positive *I. scapularis* tick was reported feeding on a human [55]. Stateside, a *B. divergens*-like agent was linked to a case of human babesiosis in Missouri [20].

We provide the first documentation of transstadial passage of *B. odocoilei* in *I. scapularis* from larvae to nymphs in Canada. Overall, we observed 25 unique events of molt metamorphosis, of which five were positive for *Babesia*, including larvae to nymphs and nymphs to adults. From one particular dual infestation on a Swainson’s Thrush, two *B. odocoilei*-positive *I. scapularis* larvae successfully molted to nymphs. These larvae may have acquired this piroplasm either by transovarial transmission or directly from the bird itself. Based on a literature search, the former is the most likely. Other *Babesia* researchers have shown that transovarial transmission occurs between *I. scapularis* females and their eggs [45]. Transovarial transmission is well-established for some *Babesia* species, such as *B. bovis* and *B. divergens,* but it is not documented for small-sized *Babesia* spp., such as *B. microti* [23,56,57]. 

While the majority of songbird-transported ticks infected with *B. odocoilei* were found during spring migration, PCR-positive ticks were also found during the summer nesting and fledging period and, likewise, during the autumn migration. Depending on bird species, autumn migration typically starts August 1st. When *I. scapularis* ticks become infected with *B. odocoilei* anytime during the four life stages (eggs, larvae, nymphs, adults), they can retain the infection throughout the life cycle. During the stage-to-stage molt, *B. odocoilei* is retained in the midgut, which stays intact, and it maintains its viability. Physiologically, *B. odocoilei* successfully moves through all four life stages (i.e., female to eggs, eggs to larvae, larvae to nymphs, and nymphs to adults) [11,45]. 

There is a paucity of information on whether any bird species may be infected with or serve as a reservoir of *B. odocoilei*. Songbirds may hold *B. odocoilei* for a short period of time in their bodies; nevertheless, without a bird–tick–pathogen study, which includes drawing blood from songbirds, we will not know the answer. In the case of week-old chickens, Lyme disease spirochetes only remain infectious for one week [22]. In order to conduct a xenodiagnostic study using an avian host and ticks, a special permit for animal research would be required. In addition, a veterinarian would be needed during the spring and fall banding period to conduct venipuncture sampling. Nonetheless, bird banders are apprehensive about putting extra stress on captured birds, especially during peak migration. 

The competence of songbirds as reservoirs of *B. odocoilei* has remained unresolved. Since none of the *H. leporispalustris*, *I. brunneus*, and *I. muris* was positive for *B. odocoilei*, these bird parasitisms add further evidence to the incompetence of songbirds as reservoirs. Although certain passerine birds may hold *B. odocoilei* in their systems, even for a short time, further enzootic bird-tick-pathogen research is required to determine the competence of selected songbirds as a reservoir for *B. odocoilei*.

The occurrence of *B. odocoilei* in *I. scapularis* nymphs collected from passerines was approximately 12.63%. Our findings are consistent with other tick researchers in the northern U.S. states who collected questing *I. scapularis* adults [49]. Since the midgut stays intact during the molt, fully engorged nymphs have equivalent infection prevalence to questing adults. Thus, when engorged *I. scapularis* nymphs molt, the infection prevalence remains the same in unfed adults. Once a motile life stage (i.e., larva, nymph, female) becomes infected with *B. odocoilei*, the infection is held throughout each life stage to the next.

Each of the five sites studied meet the criteria (six or more ticks of a motile life stage) for an established population of blacklegged ticks [58]. Whenever songbirds are nesting within an established population of *I. scapularis*, they have ample opportunity to become parasitized by ticks infected with pathogens, such as *B. odocoilei* [53]. Songbirds are incidental hosts of larval and nymphal *I. scapularis*, while white-tailed deer are resident hosts that are reservoirs of *B. odocoilei*. Coupled together, cervids, songbirds, and *I. scapularis* ticks enhance the enzootic transmission cycle of *B. odocoilei*. Transstadial passage and transovarial transmission of *B. odocoilei* in blacklegged ticks perpetuate this apicomplexan piroplasm in Canada. 

## 5. Conclusions

Songbirds play a role in the wide dispersal of *B. odocoilei*-infected blacklegged ticks. Based on our findings, we found that bird-transported *I. scapularis* nymphal ticks in Ontario and Québec have a prevalence of *B. odocoilei* infection approximating 12.63%. The prevalence of *B. odocoilei* may be as high as 16.67% in Ontario and Québec. Using molting of ticks, we provide substantial evidence that *B. odocoilei* perpetuates babesial infection through the complete life cycle of *I. scapularis*. Our research provides the first documentation of transstadial passage of *Babesia odocoilei* in *I. scapularis* from larvae to nymphs in Canada, and this study implicates transovarial transmission in the ecology of *B. odocoilei*. Notably, white-tailed deer and *I. scapularis* ticks are the key components that maintain a perpetual enzootic transmission cycle of *B. odocoilei*. Since babesial piroplasms can cause death in people, especially if they are splenectomized, immunocompromised, or have a tick-borne, zoonotic co-infection [4,59,60,61], further work is needed to determine if *B. odocoilei* is pathogenic to humans.

## Figures and Tables

**Figure 1 pathogens-09-00781-f001:**
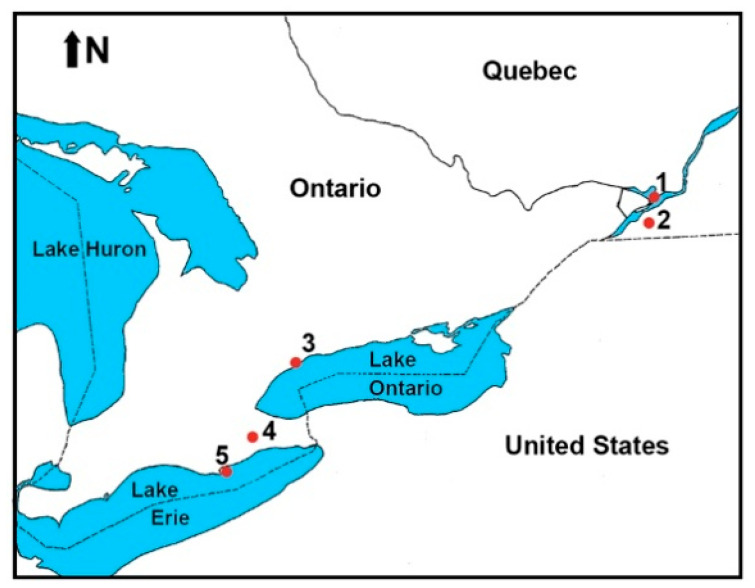
Map of Ontario and Québec showing tick collection sites. (1) McGill Bird Observatory, Ste-Anne-de-Bellevue, Québec; 45.43 N, 73.94 W. (2) Montée Biggar, Québec; 45.09 N, 74.22 W. (3) Fatal Light Awareness Program, Toronto, Ontario; 43.74 N, 79.37 W. (4) Ruthven Park National Historic Site Banding Station, Haldimand Bird Observatory, Cayuga, Ontario; 42.97 N, 79.87 W. (5) Long Point Bird Observatory, Long Point (Port Rowan), Ontario; 42.52 N, 80.17 W.

**Figure 2 pathogens-09-00781-f002:**
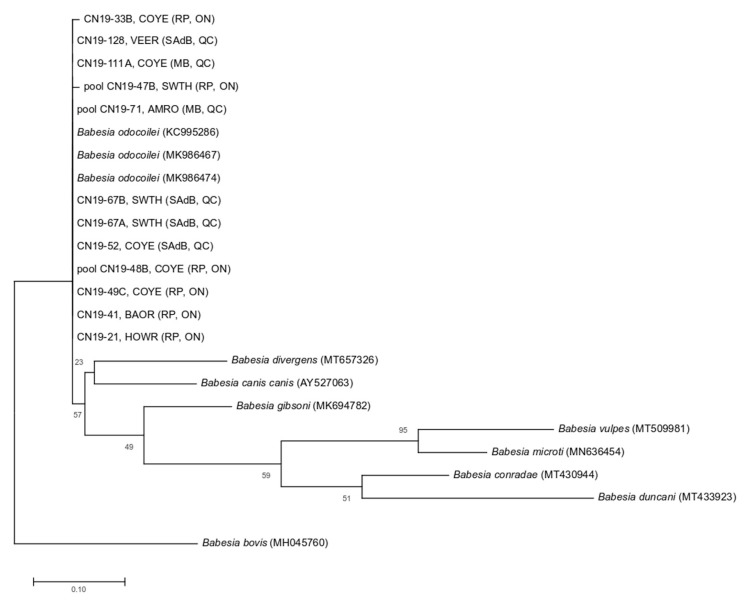
Maximum likelihood phylogenetic tree of 18S rRNA sequences from *Babesia*-positive ticks collected from songbirds in Canada during 2019, with 11 published sequences of nine different reference *Babesia* species for comparison. Labels include tick ID, bird host species, and location. Alphanumeric values in brackets denote published GenBank sequences. The scale bar represents the percentage of genetic variation along tree branches. Abbreviations: AMRO, American Robin; BAOR, Baltimore Oriole; COYE, Common Yellowthroat; HOWR, House Wren; SWTH, Swainson’s Thrush; VEER, Veery; MB, Montée Biggar (QC); SAdB, Sainte-Anne-de-Bellevue (QC); RP, Ruthven Park (ON).

**Figure 3 pathogens-09-00781-f003:**
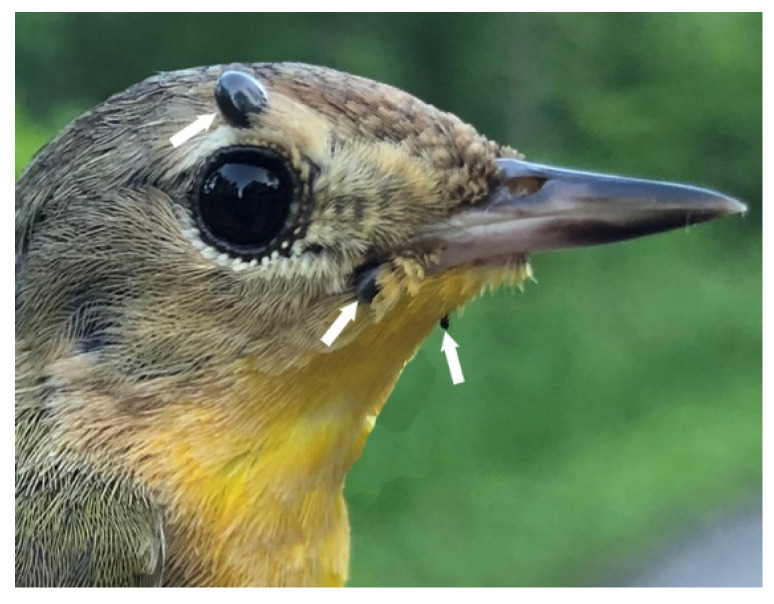
Common Yellowthroat, female, parasitized by blacklegged ticks, *Ixodes scapularis*. This ground-foraging bird species was commonly infested by bird-feeding ticks. Photo credits: Ana Morales.

**Table 1 pathogens-09-00781-t001:** Infestation of ixodid ticks collected from songbirds, and prevalence of *Babesia odocoilei* in songbird-transported ticks in Ontario and Québec, Canada, 2019. Hlp, *Haemaphysalis leporispalustris*; Ibr, *Ixodes brunneus*; Imu, *Ixodes muris*; Isc, *Ixodes scapularis*. L, larva(e); N, nymph(s); F, female(s). preval. stands for prevalence.

Bird host species	No. hosts	No. ticks	Hlp		Ibr		Imu			Isc		No. ticks/host	*Babesia* preval. (%)
			L	N	N	F	L	N	F	L	N		
American Redstart, *Setophaga ruticilla*	1	1	-	-	-	-	-	-	-	-	1 ^‡^	1.0	0
American Robin, *Turdus migratorius*	3	12	-	-	-	-	-	-	-	-	12	4.0	16.67
Baltimore Oriole, *Icterus galbula*	3	4	-	-	-	-	-	-	-	-	4 (2 ^§^)	1.3	25.00
Black-and-white Warbler, *Mniotilta varia*	1	1	-	-	-	-	1 *	-	-	-	-	1.0	0
Blue Jay, *Cyanocitta cristata*	3	4	-	-	-	1	-	-	-	-	3	1.3	0
Blue-winged Warbler, *Vermivora cyanoptera*	1	1	-	-	-	-	-	1	-	-	-	1.0	0
Canada Warbler, *Cardellina canadensis*	1	1	-	-	-	-	-	-	-	1	-	1.0	0
Carolina Wren, *Thryothorus ludovicianus*	1	1	-	-	-	-	-	1	-	-	-	1.0	0
Common Grackle, *Qiscalus quiscula*	1	1	-	-	-	-	-	-	-	-	1	1.0	0
Common Yellowthroat, *Geothlypis trichas*	24	42	1	1	-	-	1 *	-	-	-	39 (2 ^‡^, 1 ^§^)	1.8	15.63
Gray Catbird, *Dumetella carolinensis*	2	2	-	-	-	-	-	-	-	-	2	1.0	0
Hermit Thrush, *Catharus guttatus*	3	6	-	-	-	-	-	-	-	-	6 (1 ^‡^)	2.0	0
House Wren, *Troglodytes aedon*	3	6	-	-	-	-	1	2	1 ^†^	-	2	2.3	14.29
Mourning Warbler, *Geothlypis philadelphia*	5	5	-	-	-	-	-	1	-	-	4	1.0	0
Nashville Warbler, *Leiothlypis ruficapilla*	2	2	-	-	-	-	-	-	-	-	2 (1 ^‡^)	1.0	0
Northern Waterthrush, *Parkesia noveboracensis*	4	4	-	-	-	-	-	2	-	-	2	1.0	0
Ovenbird, *Seiurus aurocapilla*	1	1	-	-	-	-	-	-	-	-	1^‡^	1.0	0
Palm Warbler, *Setophoga palmarum*	1	1	-	-	-	-	-	-	-	-	1	1.0	0
Red-eyed Vireo, *Vireo olivaceus*	1	1	-	-	-	-	-	-	-	1 *	-	1.0	0
Rose-breasted Grosbeak, *Pheucticus ludovicianus*	2	2	-	-	-	-	-	-	-	-	2	1.0	0
Ruby-crowned Kinglet, *Regulus calendula*	1	1	-	-	1	-	-	-	-	-		1.0	0
Song Sparrow, *Melospiza melodia*	3	4	-	-	-	-	1 *	2	-	-	1	1.3	0
Swainson’s Thrush, *Catharus ustulatus*	6	13	-	1	-	-	2 (1 *)	1	-	2 *	7 (1 ^‡^)	2.2	30.00
Swamp Sparrow, *Melospiza georgiana*	1	1	-	-	-	-	-	1	-	-	-	1.0	0
Tennessee Warbler, *Leiothlypis peregrina*	1	1	-	-	-	-	-	1	-	-	-	1.0	0
Veery, *Catharus fuscescens*	10	27	1	2	-	-	2 *	-	-	-	22 (4 ^§^)	2.7	3.85
White-throated Sparrow, *Zonotrichia albicollis*	4	4	-	-	-	2	-	-	-	-	2 (1 ^‡^)	1.0	0
Winter Wren, *Troglodytes hiemalis*	2	5	-	-	-	-	-	-	-	5	-	2.5	0
Yellow Warbler, *Setophaga petechia*	2	3	-	-	-	-	-	-	-	-	3	1.5	0
Total	93	157	2	4	1	3	8	12	1	9	117	1.69	8.89

* collected as a larva, held to molt, tested as a nymph; ^†^ collected as a female, held to lay eggs, tested, eggs were held to hatch, tested as larvae; ^‡^ collected as a nymph, held to molt, tested as a female; ^§^ collected as a nymph, held to molt, tested as a male.

**Table 2 pathogens-09-00781-t002:** Prevalence of ixodid ticks collected from songbirds at five locations in Ontario and Québec, 2019.

Location	No. hosts	No. ticks	*Haemaphysalis leporispalustris*		*Ixodes brunneus*		*Ixodes muris*			*Ixodes scapularis*		No. ticks/host	*Babesia* preval. (%)
			L	N	N	F	L	N	F	L	N		
**Long Point (ON)**	12	16				2					14 (3 ^‡^)	1.33	0
**Ruthven Park (ON)**	25	46	1		1			2		5	37 (2 ^‡^, 3 ^§^)	1.84	16.67
**Toronto (ON)**	2	2									2 (1 ^‡^)	1	0
**Montée Biggar (QC)**	18	44	1	3							40 (4 ^§^)	2.44	5.26
**Sainte-Anne-de-Bellevue (QC)**	36	49		1		1	8 (6 *)	10	1 ^†^	4 (3 *)	24 (2 ^‡^)	1.36	8.89
**Total**	93	157	2	4	1	3	8	12	1	9	117	1.69	8.89

L, larva(e); N, nymph(s); F, female(s). * collected as a larva, held to molt, and tested as a nymph; ^†^ collected as a female, held to lay eggs, then female tested, eggs hatch, and larvae tested; ^‡^ collected as a nymph, held to molt, and tested as a female; ^§^ collected as a nymph, held to molt, and tested as a male.

**Table 3 pathogens-09-00781-t003:** Bird–tick–pathogen associations for ixodid ticks testing positive for *Babesia odocoilei* collected from songbirds in Ontario and Québec, 2019.

Sample ID	Date collected	Location	Bird species	Life stage	GenBank ID
**CN19-21**	5 May	Ruthven Park (ON)	House Wren	Nymph	MT830841
**CN19-33B**	16 May	Ruthven Park (ON)	Common Yellowthroat	Nymph to female	MT830842
**CN19-41**	18 May	Ruthven Park (ON)	Baltimore Oriole	Nymph to male	MT830843
**CN19-52**	21 May	Sainte-Anne-de-Bellevue (QC)	Common Yellowthroat	Nymph	MT830835
**CN19-48B (n = 5)**	22 May	Ruthven Park (ON)	Common Yellowthroat	Nymph	MT830846
**CN19-47B (n = 4)**	22 May	Ruthven Park (ON)	Swainson’s Thrush	Nymph	MT830845
**CN19-49C**	23 May	Ruthven Park (ON)	Common Yellowthroat	Nymph to female	MT830844
**CN19-67A**	31 May	Sainte-Anne-de-Bellevue (QC)	Swainson’s Thrush	Larva to nymph	MT830836
**CN19-67B**	31 May	Sainte-Anne-de-Bellevue (QC)	Swainson’s Thrush	Larva to nymph	MT830837
**CN19-71 (n = 7)**	07 Jun	Montée Biggar (QC)	American Robin	Nymph	MT830840
**CN19-111A**	20 Jul	Montée Biggar (QC)	Common Yellowthroat	Nymph	MT830838
**CN19-128**	20 Aug	Sainte-Anne-de-Bellevue (QC)	Veery	Nymph	MT830839

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
