# Peer review of "Detection of Babesia odocoilei in Ixodes scapularis Ticks Collected from Songbirds in Ontario and Quebec, Canada"

_pathogens, 2020, doi:10.3390/pathogens9100781_

Round 1

Reviewer 1 Report

The manuscript by Scott et al., described the Babesia odocoilei circulation in the Black-legged tick Ixodes scapualris. The manuscript is sound and bring a new data about the occurrence of the B. odocoilei in various song birds in the region of the Ontario and Québec. The authors used molecular detection of the parasite in fed stages of larvae , nymphs and adult ticks collected from 29 songbird species. For the first time the authors confirmed the transfer of this pathogen from larva to nymph ticks in Canada.  I have only minor cements for the authors that need to be addressed.    

Line 17. The genus name for the I. muris, I. brunneus should be written in full while is the first time mentioned here. The Ixodes scapularis mentioned above does not count.

Line 18-19 In the method is written 5 locations of collection .Why only 3 mentioned here? If two of them were 0% positive tick should me mentioned to avoid confusions. For example “Among 5 investigated areas only three showed positive ……as follows….”

Line 19 Please change “ticks” to “nymphs”.

Line 25 So the collected ticks were larvae or nymphs? Or some engorged larvae were collected and the once they molt they were investigated? Please clarify this.

Line 70 Not always the partially tick can molt until they reach the critical weight. Did you analyze those as well even if they did not molt?

Line 83 How did you identified the unfed ticks. You mean nonattached or molted ticks from you collection? Please clarify. The table 1 shows only testing molted (unfed) ticks. In what occasion you tested engorged ticks?  

Line 85 Why 33 ticks were pooled? For the DNA extraction? Ticks from different birds?

Line 88 Should not be 10 μM?

Line 158 What do you mean combined larvae and nymphs. Is not clear here. Did you mean The prevalence for both (larvae and nymphs together)….were ….?

The phylogenetic tree looks strange for the detected sequences. Two upper groups are mix of the identical sequences 3 for first and 12 in second group? Pleas can you clarify this? If the case there is not need to list them all. And alignment suppl. figure would be nice to show B. odocoilei identity among obtained sequences.

Author Response

Response to Reviewers' Comments: Manuscript: Pathogens-921557

Reviewer 1

L17: The full genus name has been included. Whenever a scientific name is first mentioned, we wrote the full spelling of the scientific name (both genus and species).

L18-19: The number of ticks collected at the 2 sites had a prevalence of zero due to a low number of bird-feeding ticks. This is spelled out in the text. Table 2 provides the prevalence of each of the 5 sites.

L19: Ticks have been changed to nymphs.

L25: The sentence has been re-worded. Nymphs cannot molt to larvae; this would be virtually impossible.

L70: We did not weigh ticks. The decision to have any given engorged tick molt was based on experience.

L83: The ticks were identified using the reference given. The lead author (J.D.S.) has 29 years experience in identifying ticks, and has identified 38 species of ticks collected in Canada.

L85: 33 ticks were pooled for subtle expediency which has been clarified in the text.

L88: The 100 µM has been changed to 5 µM.

L158: This sentence has been re-worded for clarity. The prevalence for both larvae and nymphs combined revealed that 12 (11.54%) of the 104 ticks were positive for B. odocoilei.

Reviewer 2 Report

I read the manuscript with great interest. The authors follow the publisher's recommendations regarding text formatting. (except for tables. Since I have a PDF version, I am not able to 100% determine it, but it seems to me that the tables are inserted photos from the source document)

L11, I suggest giving the systematic position of the studied group of birds

Research area: I suggest that throughout the manuscript authors adopt one version of the writing, Québec or Quebec

L55 give the short name of I. scapularis, the full name is written in L53

L56 similar to the L55 (with Borrelia and Babesia)

L112 listing the identified species of ticks, do it from the least numerous

Author Response

Response to Reviewers' Comments: Manuscript: Pathogens-921557

Reviewer 2

None of the tables is photos.

L11: The geographic coordinates are given in the figure caption for Figure 1.

         Quebec in the title rather than Québec makes this paper easier to cite.

L55: This has been corrected.

L56: These have been corrected.

L112: We prefer to list the tick species alphabetically.

We have added a statement about the phylogenetic tree that explains the closeness of the B. odocoilei. We believe it is important to list all B. odocoilei amplicons.